# Innate Immunity in Mucopolysaccharide Diseases

**DOI:** 10.3390/ijms23041999

**Published:** 2022-02-11

**Authors:** Oriana Mandolfo, Helen Parker, Brian Bigger

**Affiliations:** 1Division of Cell Matrix Biology and Regenerative Medicine, Faculty of Biology, Medicine and Health, University of Manchester, 3721 Stopford Building, Oxford Road, Manchester M13 9PT, UK; oriana.mandolfo@postgrad.manchester.ac.uk; 2Division of Immunology, Immunity to Infection and Respiratory Medicine, The Lydia Becker Institute of Immunology and Inflammation, Manchester Collaborative Centre for Inflammation Research, Faculty of Biology, Medicine and Health, University of Manchester, Manchester Academic Health Science Centre, Manchester M13 9PT, UK; helen.parker@manchester.ac.uk

**Keywords:** lysosomal storage disorders, mucopolysaccharidosis, glycosaminoglycans, neuropathology, neuroinflammation, IL-1

## Abstract

Mucopolysaccharidoses are rare paediatric lysosomal storage disorders, characterised by accumulation of glycosaminoglycans within lysosomes. This is caused by deficiencies in lysosomal enzymes involved in degradation of these molecules. Dependent on disease, progressive build-up of sugars may lead to musculoskeletal abnormalities and multi-organ failure, and in others, to cognitive decline, which is still a challenge for current therapies. The worsening of neuropathology, observed in patients following recovery from flu-like infections, suggests that inflammation is highly implicated in disease progression. This review provides an overview of the pathological features associated with the mucopolysaccharidoses and summarises current knowledge regarding the inflammatory responses observed in the central nervous system and periphery. We propose a model whereby progressive accumulation of glycosaminoglycans elicits an innate immune response, initiated by the Toll-like receptor 4 pathway, but also precipitated by secondary storage components. Its activation induces cells of the immune system to release pro-inflammatory cytokines, such as TNF-α and IL-1, which induce progression through chronic neuroinflammation. While TNF-α is mostly associated with bone and joint disease in mucopolysaccharidoses, increasing evidence implicates IL-1 as a main effector of innate immunity in the central nervous system. The (NOD)-like receptor protein 3 inflammasome is therefore implicated in chronic neuroinflammation and should be investigated further to identify novel anti-inflammatory treatments.

## 1. The Mucopolysaccharidoses

The mucopolysaccharidoses (MPSs) are a rare heterogeneous group of lysosomal storage disorders (LSDs) (4.5:100,000 live births) [1] which result from the defective catabolism of mucopolysaccharides, also termed glycosaminoglycans (GAGs) [2]. This is caused by mutations in the genes encoding the lysosomal enzymes responsible for the degradation of GAGs [3], leading to lysosomal and extracellular accumulation of partly degraded GAG molecules. GAGs are long-chain linear negatively-charged sugars, whose repeating disaccharide units consists of an amino sugar, either an *N*-acetyl-d-glucosamine (d-GlcNAc) or *N*-acetyl-d-galactosamine (d-GalNAc), and an uronic acid, namely d-glucuronic acid (d-GlcA), l-Iduronic acid (l-IdoA) or a galactose [4]. GAGs are normally found in proteoglycan structures, and include heparin/heparan sulphate (HS), chondroitin/dermatan sulphate (CS/DS), hyaluronan (HA) and keratan sulphate (KS) [5]. The lysosomal degradation of GAGs is a stepwise mechanism where individual sugars are removed from the end of the molecule by different enzymes. The deficiency in any of these enzymes results in eleven described mucopolysaccharide diseases (Table 1) [6].

Clinical symptoms usually manifest in patients during early childhood, although the onset and severity is quite variable amongst disease subtypes [3]. Generally, the accumulation of KS, CS and DS seems to be mostly associated with progressive skeletal dysplasia (dysostosis multiplex), cardiac conditions and hepatosplenomegaly. The subtypes caused by accumulation of HA are characterised by coarse facial features, dwarfism and soft tissue nodules surrounding the joint. Finally, the MPSs caused by incomplete HS degradation usually result in cognitive decline in patients, including abnormal behaviour and neuro-inflammation [6,7]. In this regard, the Royal Manchester Children’s Hospital, as well as other clinicians looking after similar patients, have anecdotally reported a more severe cognitive decline in MPS patients following recovery from a flu-like infection, thus suggesting the implication of inflammation in disease progression. In fact, there is evidence to suggest that the progressive accumulation of GAGs might trigger an immune response that eventually gives rise to inflammation [8,9]. At the same time, the latter might also lead to changes that eventually cause the accumulation of secondary substrates, which themselves could be immunostimulatory [9,10,11], thus establishing an inflammatory loop which leads to neurodegeneration. We have been interested in unveiling the link between the altered GAG catabolism and the inflammatory response, and only now are we able to shed further light on the immunological perspective of disease.

## 2. Pathology in MPSs

### 2.1. Primary Storage of GAGs

The first step in MPS pathology is the excessive accumulation of GAGs in cells and extracellular matrix, due to their defective degradation pathway [6]. While MPS I, II and VII are characterised by storage of both HS and DS, the latter is usually not a peculiarity of MPS III, although there is evidence of a small DS increase in MPS III patient samples and animal models [12,13]. In fact, defective enzymes in MPS I, II and VII are involved in both HS and DS catabolic pathways [14], whereas defective enzymes in MPS IIIA, IIIB, IIIC and IIID are involved in the stepwise degradation of only HS fragments [15]. A study on MPS I showed that accumulation of HS upregulates HS sulphation through a positive feedback mechanism, which could potentially worsen the neurological impairment [16]. Sulphated domains present on HS chains normally permit growth factor and protein guidance binding, thus excessive or abnormally sulphated HS might disrupt or dysregulate HS-dependent processes, such as neuronal proliferation and survival, synaptic formation and function maintenance [17,18]. The addition of sulphate groups happens via glucosaminyl *N*-deacetylase/*N*-sulfotransferase (NDST) Golgi enzymes, which direct N- and O-sulphation along the HS chains [19]. In this respect, significant increases in N-, 6-O-, and 2-O HS sulphation were detected in MPS I mouse liver, MPS I, IIIA and IIIB mouse brain [18] and in MPS I patient serum and urine [16]. Especially, increased levels of tri-sulphated disaccharide GlcA(2S)-GlcNS(6S) or IdoA(2S)-GlcNS(6S) were observed in MPS I, IIIA and IIIB mouse brains, with a consequent reduction in non- and mono-sulphated disaccharides [18]. Notably, many chemokines rely on HS as a coreceptor for binding their cognate ligands, and sulphation state is important in regulating this interaction [9,20,21]. Interestingly, some of these HS co-factors seem to prefer binding to 2-O sulphated regions. Among these factors there is the stromal cell derived factor-1 (SDF-1 or CXCL12) [18], which is involved in calcium release and chemotaxis in neutrophils, monocytes and T lymphocytes [22] via CXCR4. In this respect, it has been shown that when CXCL12 is sequestered by 2-*O* sulphated HS, haematopoietic migration in MPS I is hampered [21]. Others have proved that 2-O-sulphated HS are crucial for the high binding affinity of vascular endothelial growth factor A, IL-8 and fibroblast growth factor (FGF) [23,24]. As far as the latter is concerned, signal transduction is only possible when at least one 6-*O* sulphate group is present, thus implying a 6-*O*-sulphation-mediated FGF2/HS/FGF-receptor ternary complex generation [25]. FGF-2 regulates neural stem cell (NSC) propagation, thus abnormalities in HS may result in dysfunctional FGF-2-mediated NSC proliferation in MPS disease, and may therefore lead to impaired synapse formation and neurogenesis [26]. In addition, studies on MPS I animals revealed that 6-*O*-sulphated forms have a higher affinity binding for Wnt protein compared to its receptor Frizzled, thus resulting in blockage of signal transduction [27].

Absence of the enzymes required for GAG degradation also leads to the formation of unique non-reducing end (NRE) structures of GAG chains. In this regard, reversed phase high-performance liquid chromatography analysis revealed a specific non-reducing end disaccharide, which is uniquely present in MPS IH patients and Idua deficient mice, thus representing a possible diagnostic disease marker [16]. Furthermore, this specific disaccharide storage, together with overall HS levels, was lower in the brain than in other tissues. These findings suggest that either GAG accumulation may be a slower process in the brain or, as an alternative, that neurons present a different pathological threshold for storage [16]. Later, Lawrence et al. revealed that there are specific non-reducing end disaccharides for all the deficient enzymes in HS/DS catabolism in the mucopolysaccharide diseases [15] and proposed that these NRE GAGs were the pathological component of the GAG. Indeed, as they were able to detect NRE GAGs in plasma and urine of patients, this suggests that these GAG fragments have already been partially degraded to their current state in the lysosome, and then trafficked by an unknown mechanism to the plasma and urine, where they have the potential to interact with toll-like receptors (TLR), important for innate immunity. 

Finally, HS chain length is also important for HS facilitation of chemokine binding [20], with a minimum of between a hexamer and 18–20 mer disaccharide required to stimulate binding, depending on the context. Little is currently known about the chain length of HS accumulated in MPS diseases, but data from Parker et al. suggest that the GAG accumulated is sufficient to promote excess activation of TLR4 nonetheless [9].

Very little is known about the effects of DS accumulation in the development of MPS I and III neuropathology. It is acknowledged that IDUA deficient animals may present high levels of IdoA-containing DS [16], which is generally involved in growth factor and chemokine binding. Not only has IdoA-DS been proved to interact with FGF and specific brain molecules, but it also plays a key role in modulating neural plasticity and regeneration [28]. As far as the somatic symptoms are concerned, DS seems to be a direct mediator of articular chondrocyte apoptosis through the activation of the lipopolysaccharide (LPS) signalling pathway (TLR4, potentially TLR2/6) in several MPS diseases. These include type I, II, VI and VII, where patients all experience progressive joint stiffness [29] and progressive airway pathology and remodelling [30]. Indeed, a retrospective study of adenoid and tonsils from patients with MPS I, IIIA, IV, and VI, suggested significant basement membrane remodelling, associated with GAG storage [31], likely stimulated by an inflammatory environment. Simonaro et al. directly linked bone and joint pathology in MPS VI and VII animals to stimulation of the TLR4 pathway [8,32]. The role of KS in pathology in MPS IV is again very poorly understood and unclear. Whilst there may be accumulation of secondary storage molecules to some extent, and airway remodelling and basement membrane remodelling in adenoids and tonsils is evident [31], it is likely that KS itself is somehow also involved in the severe inflammatory responses seen in MPS IV patients [33].

### 2.2. Lysosomal Stress

By being the first site of GAG storage, the lysosomes are usually characterised by abnormal morphology [34]. Notably, lysosomal swelling and vacuolation were reported in MPS mice neurons and glia [13,18,35] as well as in MPS human neural stem cells (NSC) [36,37]. Abnormalities in lysosomal morphology could be predictive of jeopardised lysosomal membrane integrity and permeability [34]. This hypothesis is supported by a study on MPS I mice, where a lower content of H^+^ was detected in lysosomes, along with increased storage of cytosolic Ca^2+^ and cysteine proteases activity [38]. Overall, the altered pH homeostasis and molecule leakage might not only compromise the acidic organelle function, but could also be directly implicated in inflammation [34]. In particular, the reduction in H+ consequently leads to increasing pH in acidic compartments, which was proven to ultimately jeopardise the autophagosome-lysosome fusion (macroautophagy) in Chinese hamster ovary cells [39]. This fusion event is crucial to effectively achieve degradation of cellular contents and recycling of the resulting breakdown models [40]. At the same time, there is evidence that altered pH and increasing cytosolic Ca^2+^ concentration might also hamper early and late endosomes fusion, as well as late endosome and lysosome fusion [38]. This will in turn have an impact on autophagy, as recent data proved that its efficiency depends extensively on the endocytic pathway [41]. Strikingly, dysfunction in the autophagosome-lysosome fusion process was detected in MPS IIIA and MPS IH mice [36,42], thus suggesting impaired autophagy. Moreover, MPS IIIA mice also showed a lack of the lesion-associated late endosomal/lysosomal protein LIMP-2 [43].

In addition to altered lysosomal homeostasis, oxidative stress caused by mitochondrial dysfunction has also been described in nearly all the MPS diseases ([44,45,46,47]), which could contribute to worsening of inflammatory pathology through the release of reactive oxygen species (ROS) and reactive nitrogen species (NOS) [48,49]. In this respect, Hurler syndrome is marked by elevated cytochrome b558 expressing microglia, which suggests possible NADPH oxidase-dependent oxidative burst in phagocytes [49]. In MPS I and IIIA brain, as well as MPS IIIA peripheral organs, a high increase in the expression of mRNAs for oxidative stress markers was detected [46,48]. Moreover, elevated lipid peroxidation, as well as imbalanced antioxidant enzyme activity, was observed in MPS I, II and IIIB patients [44]. In particular, increased activity in superoxide dismutase and catalase enzymes was identified in multiple organs, including the brain, thus implying impaired cell homeostasis [49]. Whole transcriptome analysis of MPS II murine cerebral cortex showed upregulation of the NADPH-oxidase complex components (*Ncf1*, *Ncf4* and *Cyba*), nitric oxide synthase 3 (*Nos3*) and glutathione peroxidase 1 (*Gpx1*). Catalase (*CaT*) and superoxide dismutase (*Sod*) remained unchanged, suggesting oxidative stress dysregulation [50] in the MPS II murine model. Overall, these findings suggest existing oxidative stress in MPS diseases, although the potential causative mechanisms behind it (macrophage activation, defective autophagy and lysosomal impairment) still need to be elucidated [46]. Furthermore, increased levels of ROS have been proven to have detrimental effects on both neurons and glial cells in several MPS types [47,51].

### 2.3. Secondary Molecules

It has been hypothesised that the alteration in lysosomal pH may lead to the accumulation of secondary molecules within the lysosomes [38], which in turn aggravates impaired lysosomal homeostasis. As such, secondary accumulation of sialic acid containing glycosphingolipids, termed gangliosides, has been observed in several LSDs, including MPS I, II, III, VI and VII [18,52]. This seems to result from the inhibition of ganglioside degrading enzymes caused by the progressive accumulation of GAGs [48]. Gangliosides are normally highly expressed in the CNS and are located at the plasma membrane level [53], where they are involved in a variety of functions, such as cytokine and growth factor activity regulation, neural morphology, synaptogenesis, cell recognition and apoptosis [54]. Thus, it has been hypothesised that GM ganglioside secondary storage contributes to neurodegeneration. Interestingly, progression in GM2 ganglioside storage was detected during the first 4 months of age in MPS I and MPS III mouse models [18]. Together with proteins, sphingomyelin and cholesterol, gangliosides form lipid rafts at the surface of neurons, which play a key role in mediating signalling pathways involved in neural development and differentiation [55]. Notably, GM2 and GM3 gangliosides seem to be sequestered in vesicles, eventually forming granular cytoplasmic structures via an unknown mechanism [56].

Moreover, gangliosides are found to be sequestered in association with cholesterol in MPS I and MPS III mice [52], thus suggesting defective composition and/or turn-over of the lipid rafts, which could also elucidate neuronal dysfunction in MPS disorders [57]. The secondary storage of gangliosides observed in MPS I and MPS III mice [18] could also be related to inflammation. In fact, following a microarray-based analysis of the inflammasome related genes in GM1 gangliosidosis [58], a high expression of the inflammasome pathway was observed in GM1 gangliosidosis neural progenitor cells, with a similar pattern being witnessed in GM2 gangliosidosis. In MPSIIIA, GM2 ganglioside was found to stimulate TLR4 and precipitate a priming response, as well as an activation response when combined with GAG pre-stimulation of mixed glia [9].

Along with the accumulation of cholesterol, secondary accumulation of other classes of molecules has also been reported in several MPSs, especially in MPS I, II and III [48]. Notably, hyper-phosphorylated Tau, as well as beta amyloid (Aβ) have been detected in both MPS IIIA and IIIB mouse brain [59,60]. As far as Aβ is concerned, increased levels of neuronal HS were proven to impede brain Aβ clearance and support Aβ aggregation and amyloid plaque deposition in AD [61]. Furthermore, increased secondary storage of glypicans, (cell surface glycoproteins in which heparan sulphate glycosaminoglycan chains are covalently linked to a protein core), as well as autophagy-related proteins were observed in the medial entorhinal cortex of MPS mice, with higher levels reported in Sanfilippo mice, compared to MPS I and MPS II [60,62]. Finally, MPS IIIA mice were also marked by a secondary accumulation of α -synuclein [43], a member of the synuclein family of proteins, which are known to localise to synaptic terminals under physiological conditions [63]. Thus, its accumulation might suggest altered neurotransmission in MPS IIIIA. This is further supported by changes in pre- and post-synaptic compartments, caused by reduction in pre- and post-synaptic proteins, with no loss of synapses, in both MPS I and MPS IIIA mice [18]. 

## 3. Innate Immunity in Mucopolysaccharidoses

Acute inflammation is believed to be mediated mostly by the innate immune system [64]. This is the body’s first line of defence, with cells activated by tissue damage or pathogenic invasion. Notably, its main cellular elements include: macrophages, dendritic cells, natural killer cells, as well as epithelial cells [65]. These all display specific membrane-bound and cytosolic receptors, termed pattern recognition receptors (PRRs), which allow them to sense pathogen associated molecular patterns (PAMPs) [66] and damage-associated molecular patterns (DAMPs), ultimately activating signalling cascades that lead to the onset of adaptive immunity, phagocytosis or inflammation [65,67]. These molecular patterns include infection-related stimuli such as bacterial lipopolysaccharides (LPS) and viral components [68] as well as factors released in response to injury and cell death, including many of the storage substrates and ROS described above. At present, four classes of PRR are known: Toll-like receptors (TLRs), C-type lectin receptors, Retinoic acid-inducible gene-I-like receptors and nucleotide-binding oligomerization domain-like receptor (NLRs) [64].

### 3.1. TLR Signalling

Currently, 10 human and 12 mouse TLRs have been identified and classified into two groups based on their subcellular location: either in the plasma membrane (e.g., TLR4, which responds to LPS) or within acidified endolysosomal compartments (e.g., TLR3, which responds to viral dsRNA) [69]. TLRs have a tripartite structure which includes: a leucine-rich repeats (LRRs)-enriched ligand-binding motif, a transmembrane domain and a cytoplasmic Toll/IL-1 (TIR) receptor signalling domain [69]. Generally, the immunostimulants bind to the LRRs-enriched ligand-binding motif, thus triggering TLR activation, which eventually culminates in the TIR receptor signalling domains-mediated recruitment of the adaptor molecules myeloid differentiation primary response gene 88 (MyD88) or TIR domain–containing adapter-inducing interferon-β (TRIF) [65]. While MyD88 contributes to the activation of mitogen-activated protein kinases (MAPK), c-Jun N-terminal kinase (JNK) and NF-κB [70], eventually leading to cytokine and chemokine production, the TRIF-dependent pathway culminates in the activation of the interferon (IFN) response transcription factor IFN regulatory factor-3 and a number of IFN-β-dependent genes [71].

Interestingly, extracellular matrix components are thought to be recognised as damage-associated molecular patterns, which also bind to TLRs [72]. It has long been known that the glycosaminoglycan hyaluronan can induce TLR4-dependent inflammation [73]. In response to tissue injury, hyaluronan is degraded into small fragments which have the ability to activate innate immune cells via a TLR4 dependent mechanism in vitro and in vivo. Progressive accumulation or fragmentation of glycosaminoglycans in response to disease or injury could possibly elicit an innate immune response. There are several findings supporting the hypothesis that inflammation in mucopolysaccharidoses might be triggered through the activation of TLR4 signalling. Synovial fibroblasts obtained from MPS VI cats and rats, and MPS VII dogs showed increased expression of inflammatory molecules, including TLR4 and LPS binding protein (LBP) [74]. Moreover, TLR4 elevated expression coincided with a significant increase in NF-κB associated cytokines [74]. The same group enhanced their research by focusing on the TLR4 involvement in the pathogenesis of MPS bone and joint disease. Notably, when TLR4 was globally knocked out in MPS VII mice, restored facial and skeletal abnormalities were observed compared to normal diseased mice, as well as decreased expression of TNF-α in the serum [8]. With regard to Sanfilippo syndromes, incubation with MPS IIIB patient-derived HS oligosaccharides induced TNFα release in normal mouse microglia cultures, and higher mRNA levels of IL1ß and MIP1α. Interestingly, decreased microgliosis was observed in TLR4/MyD88 KO mice, thus suggesting HS stimulation of TLR4 and subsequent TLR4-dependent microglial priming. 

These experiments were then repeated in MPS IIIB mouse microglial cells; while higher amounts of IL1ß and MIP1α were produced in the diseased cells compared to wild-type, MPS IIIB TLR4/MyD88 KO mice showed suppression of microgliosis, with consequent reduced inflammation between 10 days and 3 months of age, with no change in the other disease markers, although inflammation re-emerged at 3 months of age [18,75]. Finally, our own research proved that TLR4 is also involved in MPS IIIA pathology. MPS IIIA mouse brains displayed increased expression of TLR4 and CD14 genes. Moreover, the treatment of WT mixed glial cultures with GAGs derived from MPS IIIA mice elicited an inflammatory response which was completely abrogated by inhibition of TLR4. However, this response was only partially reduced following neutralisation of CD14 and TLR4/MD2 [9].

There is an array of other endogenous factors which are able to interact with TLR4 and induce innate immune responses. DNA-binding protein high-mobility group box 1 (HMGB1) is released as a result of cell damage or death into the extracellular environment and has the capability of stimulating TLR4 in many neurological disorders with an inflammatory component. The levels of cytoplasmic HMGB1 are upregulated in reactive glia and spinal cord neurons in amyloid lateral sclerosis (ALS) patients [76] and similar expression of HMGB1 is observed in post-surgery hippocampal specimens from patients with drug-resistant temporal lobe epilepsy [77]. HMGB1 and TLR4 expression are significantly higher in serum from Parkinson’s disease patients, when compared to healthy volunteers; expression levels which correlate with disease severity and poor drug treatment outcomes [78]. Little evidence is available with regards to the expression patterns and role of HMGB1 in MPS. Whole transcriptome analysis of the MPS II murine midbrain was performed via RNA-seq. Pathway analysis of the autophagy and coordinated lysosomal expression and regulation (CLEAR) network showed upregulation of HMGB1, however there was an element of poor dysregulation when HMGB1 expression was analysed in other brain areas, e.g., cerebral cortex [50]. This may suggest that HMGB1 upregulation is the result of bystander effects and is likely not a main driver of inflammation in MPS. 

The role of TLR4 has been well documented in many neurodegenerative diseases. TLRs contribute to Alzheimer’s disease (AD) pathophysiology by increasing the recognition of fibrillar amyloid-beta (Aβ) and stimulating downstream signalling, as well as being implicated in the internalisation of Aβ species. As previously mentioned, deposition and accumulation of Aβ is not only observed in AD, but is also evident in both MPS patients and murine models of MPS [9,43,59,60,62,79,80,81,82]. Whole transcriptome analysis through RNA-seq revealed upregulation of *Apbb1* in MPS II murine brains, this gene encodes a protein which interacts with amyloid precursor protein (APP) and involved in AD pathogenesis.

This correlates well with what other research groups have evidenced; Ginsberg et al., found that Aβ (1–40) peptide was localised diffusely through the brains of MPS I and MPS III patients [79]. Others have also observed increased reactivity for APP, Aβ peptides and hyperphosphorylated tau in MPS models within hippocampal and cortical regions important in learning and memory [9,59,60]. Another neurodegenerative disease, Parkinson’s disease (PD) was recently linked to mutations in *NAGLU* (the enzyme deficient in MPSIIIB) [83]. The authors suggested that allelic changes in *NAGLU* increased patient susceptibility to developing Parkinson’s disease, likely due to endo-lysosomal dysfunction. In a similar fashion, whole transcriptome analysis of brains from MPS II mice showed upregulation of *Scna*, a gene involved in the production of α-synuclein [50]. Swollen interneurons within the brains of MPSIIIB patients were also positive for phosphorylated α-synuclein [80]. Beraud et al., showed that α-synuclein can directly activate microglia, inducing pro-inflammatory cytokine release and altering the expression of TLRs [84]. It is likely that accumulation of α-synuclein in MPS may not only play a role in inflammation, neurotoxicity and disease progression, but early in disease may affect TLR expression.

### 3.2. Inflammatory Cytokines

The signalling cascades triggered by the binding of TLR to an immuno-stimulant results in the activation of the macrophage itself, ultimately leading to the release of pro-inflammatory cytokines [67,85]. These are small soluble factors with pleiotropic functions, which are crucial in the up-regulation of the inflammatory response [86]. At the top of the signalling cascade two cytokines stand out: IL1 and TNF-α. While the former is involved in downstream activation of a number of other cytokines including IL-2 and IL-6 production [87], the latter is often a primary effector in bone and joint disease [88,89]. These cytokines are abundantly produced by activated peripheral leukocytes and resident CNS cells [90], and were also proved to be involved in the process of pathological pain [91]. Several groups have reported enhanced production of both peripheral and CNS pro-inflammatory cytokines in MPS diseases. As such, in MPS II and MPS IVA patients, high levels of TNF-α, IL-1β, IL-6 and/or MIP-1α were detected in both plasma and serum [91]. Moreover, increased production in liver and plasma MIP-1α, IL-1α, MCP-1, KC and RANTES were observed in MPS IIIB mice [13,35]. With regard to the CNS, the brains of untreated MPS II animals were marked by an increase in IL-1α protein, RANTES (CCL5), and monocyte chemoattractant protein (MCP-1/CCL2) [35]. Furthermore, cytometric bead array-mediated quantification of inflammatory cytokines in 9-month-old MPS I and III mouse brain showed increased levels of IL-1α, monocyte chemoattractant protein (CCL2) and MIP-1α [18], as well as both IFN-γ and its receptor [92]. Moreover, the inflammatory response obtained by stimulating murine glia with MPS III GAGs was characterised by upregulation of both TNF-α and IL-1β [9,75], which is also detected in the mouse models. Strikingly, when TLR4 was inhibited, the production of both TNF-α and IL-1β was significantly decreased, thus ultimately suggesting a potential TLR4–mediated inflammatory response in the MPS IIIA disease. 

### 3.3. The Role of Cascade Initiating Cytokines TNFα and IL-1

IL-1 is known to be implicated in a variety of functions which include: IL-2 and IL-6 production, enhancement of inflammatory reactions and development of neurodegenerative disorders [87]. Notably, both IL1α and IL1β can bind to the IL1 receptor (IL1R) family, which include: IL1R1, IL1R2 and IL1R3 [93]. While the binding of IL1R1 leads to the activation of the NF-kB, MAPK, p38 and JNK signalling pathways [87], IL1R2 acts as a decoy receptor, eventually inhibiting the activity of its ligands [94]. Conversely, IL1R3 has been proven to act as a coreceptor, by forming a trimeric complex with IL1 and IL1R1 [93]. IL1 activity is modulated by the IL-1 receptor antagonist (IL1-Ra), which competes for the IL1R1 binding site, eventually blocking IL-1-mediated cellular changes [95]. The balance between IL1 and IL1Ra has been proved to be crucial to maintain normal physiological conditions. In this regard, upon injection of LPS into healthy individuals, a first IL-1 peak was observed after 2 h, followed by an IL1-Ra peak between 3 to 6 h marked by 80-fold levels over IL-1 [96]. In fact, because of the spare receptor effect, very high levels of IL1Ra are required to achieve complete inhibition of IL-1 activity. Both IL1 and IL1RA are elevated in MPSIIIA (patients and mouse models) as well as several other MPS diseases and chronic inflammation appears to negatively impair the negative feedback loop (Figure 1).

A lentiviral-mediated haematopoietic stem cell (HSC) gene therapy approach was recently devised to target the IL-1 immune response and to investigate the effects of its abrogation on MPS IIIA neuroinflammation. Notably, MPS IIIA mouse HSCs were transfected with two different doses of lentiviral vectors expressing IL-1 receptor antagonist (IL-1Ra). Interestingly, lower expression of IL-1Ra (but still above physiological levels) resulted in the attenuation of IL-1 immune response, which in turn led to reduced microgliosis and astrogliosis in the CNS, as well as complete behavioural correction [9]. At the same time, the generation of MPSIIIA/IL-1R1^−/−^ mice, to further investigate IL-1 signalling, also resulted in reduced brain glial activation, reversal of working memory deficits and normalisation of hyperactivity. Overall, while IL-1 seems to be the main effector of innate immunity in MPS IIIA disease, cognitive functioning relies on the correct balance between IL-1 and its antagonist.

While all the previous data convey the idea that IL1 mostly drives inflammation in the CNS, TNF-α seems to be primarily involved in the periphery. By binding to the TNFR1 receptor, TNF-α mediates both pro-inflammatory and programmed cell death pathways [97]. In particular, TNF-α is known to induce neutrophil recruitment, endothelial permeability and the production of prostaglandin E2, hence determining inflammatory pain. In this respect, increased levels of TNF-α are associated with severe physical pain and impaired physical function in MPS I, II and VI patients [98]. Additionally, analysis of synovial fibroblasts and fluid from MPS VI rats, cats and dogs showed a 50-fold increase in TNF-α [74]. Based on these findings, some anti-TNF-α approaches have been used to treat the musculoskeletal symptoms in MPS VI rats [8,99]. Notably, not only was the apoptosis of articular chondrocytes reduced, but overall bone morphology and motor activity was ameliorated. Overall, these findings suggest the implication of the TLR4/TNF-α signalling pathway in MPS bone and joint disease, and the TLR4/IL-1 in neurodegeneration.

### 3.4. The NLRP3 Inflammasome and IL-1

Full activation of innate immune responses requires both a priming response and a boost in which disease-specific substrates such as HS are essential for priming an IL-1β response via TLR complexes, but other receptors/immune complexes, such as the inflammasome, must play a role in boosting the inflammatory response by triggering IL-1β maturation and secretion. MPS IIIA GAG on its own does not seem sufficient to elicit IL-1β secretion in vitro, as only a TLR4-dependent intracellular production of IL1β is observed in response to the priming stimulus, and a second stimulus is required to elicit IL-1β secretion [9]. Consistent with this, increased levels of IL-1β and IL-1Ra are detected in MPS III patient plasma and CSF, and murine plasma and organs, indicating that IL-1 is released from immune cells into its extracellular environment [9]. Inflammasomes are known to activate inflammatory caspases, which ultimately promote the maturation of both IL-1β and IL-18 [100]. They are cytosolic protein complexes which were first described as signalling platforms containing a NLR [101]. Disturbed homeostasis derived from cellular damage or microbial infection can be sensed by inflammasomes, which respond by activating pro-inflammatory cytokines to engage both the innate and adaptive immune response [102]. In response to several pathogen-derived danger signals, canonical inflammasomes such as NLRP1, NLRP3, NLRC4, proteins absent in melanoma 2 (AIMS) and pyrin are assembled by the protein pyrin or by members of the NLR and HIN domain-containing (PYHIN) protein families and culminate in the activation of caspase-1. Non-canonical inflammasomes are activated by LPS and trigger the activation of caspases 4, 5 and 11 [100]. To date, NLRP3 has been the most widely-characterised inflammasome [103], although the mechanisms regulating its activation still remain elusive [102], they include lysosomal disruption, hence its importance in lysosomal disease.

The NLRP3 inflammasome has a tripartite structure formed by: a carboxy-terminal LRR (sensor molecule), a nucleotide-binding-and-oligomerization domain, and an amino-terminal pyrin domain (PYD) [102,103]. Following activation, NLRP3 monomers undergo oligomerisation and interact with the adaptor apoptosis-associated speck-like protein containing a C-terminal caspase recruitment domain (ASC) via PYDV [101]. Subsequently, ASC self-associates and recruits pro-caspase-1, which is eventually activated through proximity-induced autocatalytic activation [103], leading to IL-1β and IL-18 maturation and pyroptosis [102]. Several studies have shown that NLRP3 activation requires at least two steps, termed as priming and assembly [104]. Generally, priming involves the engagement of PRRs (such as TLRs), which culminates in NF-kB activation, eventually leading to NLRP3 transcription [105]. Furthermore, it seems that PRR signalling also triggers some specific post-translational modifications in NLRP3 complex. Mainly, JNK1-mediated NLRP3 S194 phosphorylation has been proven to be pivotal for subsequent inflammasome K48-linked deubiquitination, which allows NLRP3 homo-oligomerisation. Secondary stimulatory signals then lead to cellular events such as ROS generation, potassium efflux, lysosome rupture or mitochondrial dysfunction, which eventually trigger inflammasome assembly, as extensively reviewed by Kelley et al. [106]. While the role of these events still remains controversial, K^+^ efflux has proved to be a central NLRP3 activator [102]. In fact, not only has it been proved that decrease in cytosolic K^+^ concentration is sufficient to activate NLRP3 [107], but most of the activation models which have been proposed converge on the potassium step. For instance, NLRP3 is known to be activated by plasma membrane channels or ionophores which all proved to render the membrane permeable to potassium [102]. Moreover, a recent study suggests that damaged lysosomal compartments lead to accumulation of multiple cytosolic cathepsins, which induce specific changes in plasma membrane potassium and calcium fluxes, hence leading to NLRP3-mediated inflammation [108]. In this respect, cathepsins have been proved to activate the NLRP3 inflammasome in Alzheimer’s disease microglia [109]. Conversely, an association between disrupted autophagy and NLRP3 activation has been observed in Gaucher disease macrophages [110]. This seems to be due to the incapability of the damaged lysosomes to correctly fuse with phagosomes, engulf ubiquitinated inflammasomes and terminate their activity. Altogether, not only could this happen in MPS diseases, but there is a chance that TLR4-signaling itself might mediate NLRP3 priming, with the secondary stimulatory signals determining its full activation. In MPS IIIA brain tissue, increased expression of subunits, activators and downstream mediators of the NLRP3 inflammasome were detected when compared to WT [9]. Interestingly, stimulation of WT-mixed glia with secondary storage substrates such as ATP, cholesterol or amyloid-β led to production of intracellular IL-1-β (a priming response only), but only elicited significant IL-1-β release (inflammasome activation), when cells were initially primed with MPS IIIA GAG, thus suggesting that both primary and secondary storage substrates are essential for inflammasome activation and IL-1β secretion in MPS diseases. 

In addition, when repeating these experiments in *Nlrp3*^−/−^ mixed glial cultures, lacking this major inflammasome component, no secretion of IL-1β was detected, hence confirming the involvement of NLRP3 inflammasome activation in this process. Overall, these findings suggest HS/TLR4/IL1-signaling-depedent priming, and secondary storage material-mediated amplification of neuroinflammation via the NLRP3 inflammasome. 

The HS/TLR4/IL1-signaling-depedent priming had also been proved in MPS IIIB, where the GAG-dependent immune responses were dramatically reduced in TLR4 and MyD88 knockout cells [75]. Nevertheless, although initial lack of microglial priming was observed in TLR4 and MyD88-deficient MPS IIIB mice, the onset of disease marker expression returned after 3 months of age and similar levels of neurodegeneration were detected. This could be explained by the fact that HS and secondary storage substrates accumulation and autophagic dysfunction may still trigger other inflammatory pathways independent of TLR4, including other TLRs and the inflammasomes, eventually culminating in the prime-boost model which was just described. Finally, the bone and joint disease correction and lack of secondary storage-mediated amplification following TLR4 knock-out in MPS VII mice further proves the pivotal role of TNF-α in the periphery, as opposed to IL-1β [8]. 

### 3.5. Cell Death

As previously mentioned, the NLRP3 inflammasome-mediated maturation of IL-1β and IL-18 leads to pyroptosis [102], namely the inflammatory process of caspase 1-dependent programmed cell death. This process is marked by an increase in the cellular size, DNA cleavage and pore formation in the plasma membrane, thus leading to rupture of the latter and subsequent release of the pro-inflammatory cellular constituents into the interstitial tissue [111]. In Alzheimer’s disease (AD) microglia, it was observed that oxidative stress is crucial in the activation of cathepsin B, a cysteine protease which is believed to play a role in both NLRP3 inflammasome activation and neuronal death [109]. In this respect, a recent study revealed elevated levels of cathepsins B in MPS IIIA mice [46]. Additionally, both in vitro and in vivo studies on microglial-secreted cathepsin B [18,112] highlighted the potential implication of the molecule in the loss of neurons and nervous system atrophy observed in MPS III. Conversely, although increased expression of both cathepsins B and D was detected in MPS I mouse hippocampus, no major neuronal loss is generally observed in MPS I mice [18,113]. In this regard, it could be possible that the proteolytic nature of these proteins may contribute instead to neuronal dysfunction and altered synaptic plasticity [113]. Furthermore, increased cathepsin B activity and leakage into the cytoplasm was also detected in MPS II neurons, along with increased IL-1β and caspase-1 activity [114]. Although both caspase-mediated, pyroptosis differs quite substantially from apoptosis. In fact, the latter is characterised by cell shrinkage, chromatin condensation and disintegration of the nucleus, which eventually leads to plasma membrane blebbing and fragmentation into membrane-bound apoptotic bodies. As these are quickly phagocytosed by macrophages, no inflammatory response is triggered [115]. In MPSs, reactive microglial release of TNF-α and MIP was proven to drive mitochondrial damage in MPS IIIC mice, which in turn mediates neuronal death [62]. In fact, TNF-α is known to induce two differential caspase-8 activation pathways that are modulated by different anti-apoptotic proteins [116]. In this respect, in MPS VI rats and cats, the higher presence of nitric oxide and TNF-α in the joints was associated with enhanced apoptosis of articular chondrocytes [117]. Microglial ROS release might also be associated with apoptosis. In fact, ROS seems to be responsible for the progressive loss of Purkinje cells, which is experienced in IDUA-knockout mouse models at 19 weeks of age [51]. Notably, periodic acid-Schiff staining unveiled accumulating material in the Purkinje cells of MPS I mice cerebellum, which is likely to cause lysosomal instability and oxidative damage in these cells [51]. In MPS II NSCs, oxidative damage with metabolic dysfunction was proved to trigger astroglial degeneration, eventually leading to reduced synaptic density and apoptosis of neurons. This model was further confirmed when low oxygen conditions and vitamin E treatment reversed the astrocytes phenotype [47]. It was observed that severe hypoxic conditions can also induce necrosis, namely cell death caused by physiochemical insults [118]. In particular, following hypoxic condition-induced necrosis, IL-1α was proven to be released into the extracellular space, eventually attracting monocytes through other chemokines [93]. While necrosis was initially believed to be accidental, today it is thought to be a regulated process (necroptosis), characterised by cellular leakage, cytoplasmic granulation and organelle and/or cellular swelling [119]. Notably, this process is initiated through the TNFα signalling pathway and is dependent on the activity of receptor-interacting protein kinase 1 (RIPK1) and RIPK3 [120]. Interestingly, recent studies demonstrated the key role of RIPK3 as activator of necrotic cell death in Gaucher’s disease (GD) neurons [121]. Notably, GD RIPK3 deficient mice showed reduced neuronal and hepatic injury, as well as enhanced survival and motor coordination. However, RIPK3 also seems to drive inflammation through several necroptosis-independent functions; as such, the absence of RIPK3 was discovered to suppress the TLR4-induced IL1β production [122]. Thus, a potential interaction between the kinase and the NLRP3 inflammasome is hypothesised.

### 3.6. Innate Immunity in Mucopolysaccharide Diseases

Overall, we are now able to propose a two-step model for innate immune system activation that is probably similar across at least the MPS diseases, with MPSIIIA providing the bulk of evidence, but possibly more widely across other lysosomal diseases. In the first step, accumulation of GAG leads to priming via TLR4. Pathogenic GAG is highly sulphated, which seems to lower the priming threshold of TLR4. It is worth noting that multiple products could induce priming—including TNFα, but the continued presence of GAG makes it a likely candidate for the continued priming of the innate system. As disease progresses, a number of factors occur including lysosomal enlargement, secondary storage, defective autophagy, mitochondrial dysfunction and reactive oxygen species production, defective calcium homeostasis, lysosomal leakage and release of cathepsin B. Several of these factors are able to directly induce the second step of NLRP3 inflammasome activation, whilst yet others (GM gangliosides, and even adventitious infection) may result in activation of other TLR pathways, thus providing further priming and/or activation. The net result is production (Priming step) and release via caspase 1 cleavage (Activation step) of mature IL1β. This interacts with IL1R1—present on neurons and microglia, and precipitates multiple downstream events, including pyroptosis (caspase 3 mediated cell death) (Figure 2).

## 4. Anti-Inflammatory Therapies in MPS Diseases

### 4.1. CNS-Targeting Therapies

Current approved therapies for MPS diseases, mainly based on enzyme replacement therapy, bring very little benefit to neurocognitive deficits, and the abrogation of neuroinflammation still remains a challenge. In this regard, substrate reduction therapy, which aims at reducing GAGs synthesis, appears to be a promising option [123]. In fact, administration of Genistein, an isoflavone found in soy products, not only led to reduced levels of HS in MPS IIIB mice, but also noticeable reduction in astrogliosis and microgliosis within the cerebral cortex [124]. Furthermore, a one-year pilot study proved that oral administration of this molecule results in improved cognitive function in MPS IIIA and IIIB patients [125]. Sadly, the phase III trial in 20 MPSIII patients proved that genistein was ineffectual at reducing GAG content in the brain, and despite modest peripheral reduction in GAG, no neurocognitive benefit was observed [126]. Alternatives that are effective at reducing GAG or GAG sulphation would be of interest.

Following identification of over-expressed inflammation-related and oxidative stress-related genes in the MPS IIIA mouse CNS, an aspirin treatment was carried out, and the genes were used as markers of neuroinflammation [46]. The treatment resulted in normalisation of inflammatory mRNA levels in MPS IIIA mouse brains, and reduction of oxidative stress also in the peripheral organs. Conversely, the corticosteroid Prednisolone was tested in MPS IIIB mice [127]. A striking decrease in astrogliosis, as well as improved long-term memory, was observed in three- to five-week-old MPS IIIB mice, following daily administration of prednisolone for six months. A recent study showed no reduction in neuropathology following prednisolone treatment in 6-week to 8-month-old MPS IIIB mice, yet correction of hyperactive behaviour and reduction in peripheral cytokines was achieved [13]. Overall, these findings suggest that a reduction in peripheral inflammation with steroid treatment might be sufficient to alleviate the behavioural abnormalities in MPS mice. While cytokines are known to be able to cross the blood brain barrier, their major site of production is in the periphery; it might be surmised that reduction in the peripheral cytokine burden could impact the central burden in the brain. Prednisolone or other corticosteroids are not, however, a great option for patients, due to their long-term side effects.

At present, targeting IL-1β with IL-1β antibodies or recombinant IL-1β receptor antagonist represents the main therapeutic approach for NLRP3-related diseases; however, it is important to state that this strategy does not guarantee full inhibition of the immune response, as this can be further amplified by other cytokines, such as IL-18. As previously mentioned, Parker et al. targeted IL-1β by transfecting MPS IIIA mouse HSCs with lentiviral vectors expressing the IL-1β receptor antagonist, IL-1Ra and also by generating an MPSIIIA × IL-1R1^−/−^ mouse model to knock out the IL1 receptor [9]. In both cases, blockage of IL-1β action led to complete behavioural correction, although only a partial reduction was observed in microgliosis, astrogliosis and no changes in lysosomal swelling were observed, suggesting that the burden of disease was still present, but the link to pathology had been broken. This is probably due to the fact that this therapeutic approach did not address the primary GAG storage and secondary substrate accumulation, which has the potential to trigger several other inflammatory pathways, if left unresolved. 

Following these findings, a phase II/III clinical trial was recently developed for MPS III patients, based on Anakinra (interleukin-1 receptor antagonist) injections [128]. While this treatment aims to improve behaviour, sleep and seizure frequency, one of its downsides resides in the discontinuous delivery of the drug. In fact, by being an immunosuppressive medicine, it could be detrimental if administered to patients with ongoing infections, which are known to be recurrent in MPS III [129] and these factors should be carefully weighed against possible benefits, ideally in a randomized controlled trial. 

Targeting IL-1β has also a potential for untargeted immunosuppression, as this cytokine can be produced by other inflammasomes, as well as other inflammatory pathways [130]. In fact, by being a potent stimulator of haematopoiesis and an accessory signal for lymphocyte activation, inhibition of this cytokine might have detrimental effect on both innate and adaptive immune responses, hence leaving the patient with a dysfunctional defence line against infection or injury [131]. For these reasons, pharmacological inhibitors of NLRP3 inflammasome might determine a more specific and robust abolition of neuroinflammation in MPS diseases. As such, MCC950, a small-molecule inhibitor of NLRP3 inflammasome proved to efficiently block both canonical and noncanonical NLRP3 activation in human and mouse macrophages [132]. In particular, the molecule induces reduction in IL1β by preventing NLRP3-induced ASC oligomerisation, but it does not interfere with the activation of other inflammasomes, such as NLRP1 or NLRC4. Another possible candidate could be the anti-allergic drug Tranilast, which was discovered to inhibit NLRP3-NLRP3 interaction and ASC oligomerisation by binding to the NACHT domain of NLRP3 [133]. Hence, the drug acts via an ATPase independent manner and does not intervene in the upstream signalling pathway of the inflammasome. Moreover, Tranilast was further tested in several mouse models of NLRP3-related diseases and proved to have beneficial effects on the clinical phenotype [130].

### 4.2. Periphery-Targeting Therapies

Another potential therapeutic target is the TLR4-TNFα pathway. As previously mentioned, Simonaro and their group proved that anti-TNFα drugs can improve musculoskeletal manifestations in MPS diseases [8]. In addition, combining ERT with Infliximab-based therapy induced ameliorated motor function in MPS VI rats [99]. Based on these findings, clinical trials are being conducted in MPS patients, mostly addressing the treatment of joint and bone symptoms [134]. Both infliximab and adalimumab, targeting TNFα, are potential candidates for bone and joint disease.

A similar approach was performed by administering pentosan polysulfate (PPS), which is a TNFα antagonist and HS mimic [72]. This anti-inflammatory drug not only led to improved clinical outcome in MPS VI rats after 8-month treatment [135], but also reduction in GAG storage in urine. In addition, decreased pro-inflammatory cytokines levels were found in the cerebrospinal fluid of MPS I dogs, following subcutaneous administration [136]. Notably, clinical trials in MPS I patients resulted in drug toleration and improved bone and joint symptoms [137]. All in all, PPS might be a potential drug for the treatment of neural inflammation in MPS I and MPS IIIA diseases; in this regards, future studies should analyse the clinical outcomes related to the CNS injection of the drug or enhance its BBB permeable properties; in fact, the size of the molecule might limit its passage across the BBB. Nevertheless, a central effect might be obtained if significant reduction of inflammation is achieved in the periphery. 

## 5. Conclusions

We propose that the progressive accumulation of abnormal GAGs triggers an inflammatory response in MPSs, which is initiated by TLR4 activation (Figure 2). Altered highly sulphated GAG fragments are recognized as DAMPs, eventually triggering several signalling cascades which result in the activation of macrophages and microglia. The latter, in turn, releases a number of pro-inflammatory cytokines, such as TNF-α and IL-1, which induce progression through chronic neuroinflammation. While TNF-α is mostly associated with bone and joint disease in MPSs, IL-1 seems to be more involved in CNS inflammation. In this regard, our extensive review of disease pathology seems to suggest TLR4/IL1-signaling-depedent priming, and secondary storage material-mediated amplification of neuroinflammation. In fact, the priming results in the activation of the NF-Kβ pathway, which could also subsequently prime the NLRP3 inflammasome assembly. Simultaneously, the progressive accumulation of primary and secondary substrates within the lysosome could eventually hamper a multitude of key functions, ultimately contributing to the full activation of the inflammasome (Figure 2). Once activated, the latter induces IL-1β and IL-18 maturation and pyroptosis. Despite the presence of anti-inflammatory factors, such as IL10, IL1Ra, seen in many chronic conditions, we hypothesise that the negative feedback loop of IL1beta/IL1Ra is compromised by continuous production of proinflammatory cytokines. In all, not only is there significant evidence suggesting the involvement of the NLRP3 inflammasome in the chronic neuroinflammation observed in MPS diseases, but we propose IL-1 as a main effector of the pathological effects of innate immunity in the neuropathological MPSs, including manifestation of altered behaviours.

## Figures and Tables

**Figure 1 ijms-23-01999-f001:**
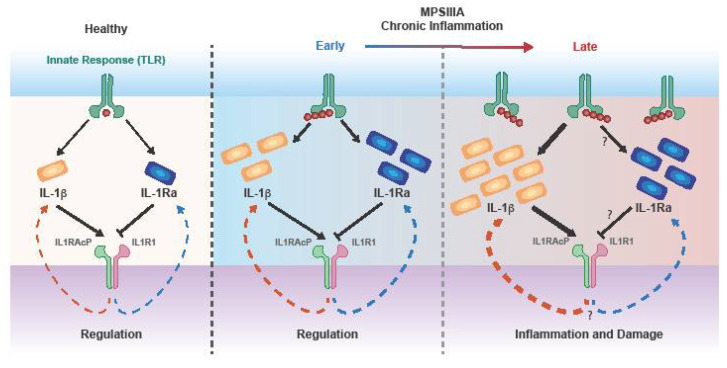
IL-1 feedback. Under normal conditions, TLR signalling leads to IL-1β production. The cytokine binds to IL-1R1 which associates with the coreceptor IL1RAcP to form the high affinity interleukin-1 receptor complex. IL1 activity is modulated by the IL1-Ra, which competes for the IL1R1 binding site, preventing association with IL1RAP to form a signalling complex. Due to the spare receptor effect, higher levels of IL1-Ra are required to fully inhibit IL-1. In MPS IIIA, GAG-induced TLR4 signalling results in increased production of both cytokines. However, with disease progression, the exacerbated production of IL-1 cannot be antagonized by adequately high levels of IL-1Ra, thus determining an imbalance in the negative-feedback loop.

**Figure 2 ijms-23-01999-f002:**
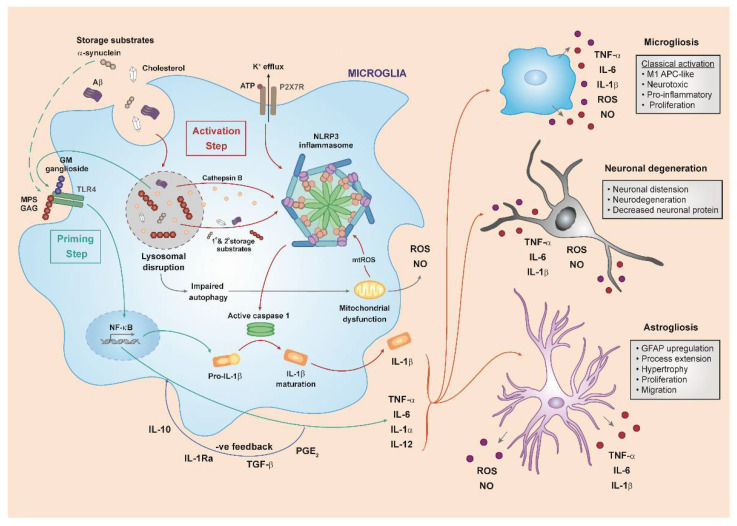
Two-step model for innate immune system activation in MPS diseases. The lack of lysosomal enzyme activity associated with MPS leads to the accumulation of highly sulphated GAG that could either be exocytosed and bind to TLR4 or released intracellularly due to lysosomal disruption. Secondary storage substrates released from cells may also activate TLR4—GM2 ganglioside for example. TLR4 signalling culminates in the activation of the NF-Kb pathway, which promotes pro-IL-1β and NLRP3 transcription (Priming step), as well as transcriptional induction of other pro-inflammatory cytokines. Many of the pathological events associated with disease progression, such as defective autophagy, lysosomal leakage and cathepsin release, mitochondrial dysfunction, reactive oxygen species, and changes in potassium permeability, can further boost inflammation by triggering the NLRP3 inflammasome assembly and activation (Activation step). This eventually leads to secretion of mature IL-1β via caspase-1 cleavage which, together with the other pro-inflammatory cytokines, causes multiple downstream events, such as microglia and astrocyte activation and neuronal degeneration, including pyroptosis.

**Table 1 ijms-23-01999-t001:** Classification of MPS diseases.

Category	Disease	Deficient Enzyme	Accumulating Substance	Gene
MPS I-S	Scheie		DS, HS	*IDUA*
MPS I-HS	Hurler-Scheie	α-l-iduronidase		
MPS I-H	Hurler			
MPS II	Hunter	Iduronate 2-sulfatase	DS, HS	*IDS*
MPS IIIA	Sanfilippo	Heparan *N*-Sulfatase	HS	*SGSH*
MPS IIIB		α-*N*-acetylglucosaminidase Acetyl-CoA-α		*NAGLU*
MPS IIIC		glucosaminidase		*HGSNAT*
MPS IIID		*N*-acetylglucosamine-6-sufatase		*GNS*
MPS IVA	Morquio A	Galactosamine-6-sulfatase	KS, CS	*GALNS*
MPS IVB	Morquio B	Β-galactosidase	KS	*GBL1*
MPS VI	Maroteaux-Lamy	*N*-acetylgalactosamine-4-sulfatase	DS	*ARSB*
MPS VII	Sly	β-Glucuronidase	DS, HS, CS	*GUSB*
MPS IX	Natowicz	Hyaluronidase	HA	*HYAL1*

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
