# Peer review of "Innate Immunity in Mucopolysaccharide Diseases"

_ijms, 2022, doi:10.3390/ijms23041999_

Round 1

Reviewer 1 Report

The authors presented a very extensive review on MPS pathology with an accurate description of disregulated molecular pathways affecting immunological response. The paper is well written and accurate, also addressing therapeutic interventions, and as such I believe is acceptable in the present form. The only remark I  could make is that the complexity of the pathways described in the different section is not adequately reported graphically, in that some additional diagrams might help the reader in following the complex biochemical and cellular alterations described in MPS. Moreover there is no reference to the figures in the text.

Author Response

We thank the reviewers for their perspicacious comments and have adopted all of them

R1

The authors presented a very extensive review on MPS pathology with an accurate description of disregulated molecular pathways affecting immunological response. The paper is well written and accurate, also addressing therapeutic interventions, and as such I believe is acceptable in the present form. The only remark I  could make is that the complexity of the pathways described in the different section is not adequately reported graphically, in that some additional diagrams might help the reader in following the complex biochemical and cellular alterations described in MPS. Moreover there is no reference to the figures in the text.

We have updated figure 1 to include a little more information on mitochondrial dysfunction, the role of secondary storage molecules, defective autophagy and stimulation of TLR4 by GM2 ganglioside. We have also added further references to the figures in the text (each is already referenced once.)

Reviewer 2 Report

In the present review the authors nicely describe the different mucopolysaccharidoses, their pathologies, lysosomal stress associated with them, possible significance of secondary molecules and then concentrate on the involvement of innate immunity in this group of diseases.

The review should be accepted. I advise the authors to pay attention to the following comments: 

Page 3, lines 94-96: The authors describe that: “Since FGF-2 regulates neural stem cells (NSC) propagation, abnormalities in HS may result in dysfunctional FGF-2-mediated proliferation of MPS multipotent progenitors, and possibly to impaired synapse formation and neurogenesis [26]”. Please edit the sentence.

Page 4, lines 152-153: It is stated that: ”This fusion event is crucial to effectively achieve  degradation of cellular contents and recycling [40].”Recommended to add: …degradation of cellular contents and recycling of the resulting breakdown molecules.

Page 5, lines 203-204: “a high expression of the inflammasome pathway was observed in GM1-NPC, with a similar pattern been witnessed in GM2.”. It seems GM1 and GM2 are the gangliosides, what is NPC, Niemann-Pick disease type C? Please specify what are GM1-NPC and GM2.

Page 5, line 213: Glypicans: I suggest adding a sentence explaining that glypicans are a group of cell surface glycoproteins in which heparan sulfate glycosaminoglycan chains are covalently linked to a protein core.

Page 7, lines 290-292: “HMGB1 and TLR4 expression are significantly higher in serum from Parkinson’s disease patient, when compared to healthy volunteers; expression levels which correlated with disease severity and poor drug treatment outcomes [79]”. Please use the same tense throughout the sentence.

Figure 1:  I guess the paragraph that follows the figure is its legend, however, it is not clear. The same is true for the legend of figure 2.

Author Response

We thank the reviewers for their perspicacious comments and have adopted all of them

R2

In the present review the authors nicely describe the different mucopolysaccharidoses, their pathologies, lysosomal stress associated with them, possible significance of secondary molecules and then concentrate on the involvement of innate immunity in this group of diseases.

The review should be accepted. I advise the authors to pay attention to the following comments: 

Page 3, lines 94-96: The authors describe that: “Since FGF-2 regulates neural stem cells (NSC) propagation, abnormalities in HS may result in dysfunctional FGF-2-mediated proliferation of MPS multipotent progenitors, and possibly to impaired synapse formation and neurogenesis [26]”. Please edit the sentence.

edited as follows: FGF-2 regulates neural stem cell (NSC) propagation, thus abnormalities in HS may result in dysfunctional FGF-2-mediated NSC proliferation in MPS disease, and may therefore lead to impaired synapse formation and neurogenesis

Page 4, lines 152-153: It is stated that: ”This fusion event is crucial to effectively achieve  degradation of cellular contents and recycling [40].”Recommended to add: …degradation of cellular contents and recycling of the resulting breakdown molecules.

Line amended accordingly

Page 5, lines 203-204: “a high expression of the inflammasome pathway was observed in GM1-NPC, with a similar pattern been witnessed in GM2.”. It seems GM1 and GM2 are the gangliosides, what is NPC, Niemann-Pick disease type C? Please specify what are GM1-NPC and GM2.

NPC is actually a neural progenitor cell but we can see how this is confusing for the audience (NPC also = Niemann pick C disease). We have therefore spelt out all instances of NPC and changed the text as follows:

In fact, following a microarray-based analysis of the inflammasome related genes in GM1 gangliosidosis [58], a high expression of the inflammasome pathway was observed in GM1 gangliosidosis neural progenitor cells, with a similar pattern been witnessed in GM2 gangliosidosis. In MPSIIIA, GM2 ganglioside was found to stimulate TLR4 and precipitate a priming response, as well as an activation response when combined with GAG pre-stimulation of mixed glia

Page 5, line 213: Glypicans: I suggest adding a sentence explaining that glypicans are a group of cell surface glycoproteins in which heparan sulfate glycosaminoglycan chains are covalently linked to a protein core.

Amended as suggested

Page 7, lines 290-292: “HMGB1 and TLR4 expression are significantly higher in serum from Parkinson’s disease patient, when compared to healthy volunteers; expression levels which correlated with disease severity and poor drug treatment outcomes [79]”. Please use the same tense throughout the sentence.

Amended as suggested

Figure 1:  I guess the paragraph that follows the figure is its legend, however, it is not clear. The same is true for the legend of figure 2.

Legends for both figures have now been more clearly defined using italics